## 1 Detection of Land Subsidence due to Excessive Groundwater Use Varying

# 2 with Different Land Cover Types in Quetta valley, Pakistan Using ESA-

- **3 Sentinel Satellite Data**
- 4 Waqas Ahmad<sup>1</sup>, Minha Choi<sup>2</sup>, Soohyun Kim<sup>1,3</sup> and Dongkyun Kim<sup>1</sup>
- 5 <sup>1</sup>Department of Civil Engineering, Hongik University, Mapo-gu, Seoul, 04066 Korea
- <sup>2</sup>Water Resources and Remote Sensing Laboratory, Department of Water Resources,
- Graduate School of Water Resources, Sungkyunkwan University, Suwon, 440-746, Korea
- <sup>3</sup> National Disaster Management Institute, Jongga-ro, Jung-gu, Ulsan, Korea
- Corresponding to: Dongkyun Kim (kim.dongkyun@hongik.ac.kr)
- Abstract. In this study, we show that land subsidence in Quetta valley, Pakistan is caused by exploitation of groundwater based
- on the analysis of European Space Agency (ESA) Sentinel satellite data. For this, we performed interferometry analysis using
- twenty nine Sentinel-1 SAR images to obtain twenty eight interferograms of land subsidence for the period between 16 Oct 2014
- and 06 Oct 2016. Then, the land subsidence was compared with the land cover of the study area that was derived from Sentienl-2
- multispectral images. The results reveal that, for the period of two years, the entire study area experienced highly uneven land
- subsidence with its magnitude ranging between 10 mm to 280 mm. The spatial pattern of land subsidence showed a high
- correlation with that of land covers. While urban and cultivated area with high groundwater extraction showed great amount of
- land subsidence, barren and seasonally cultivated area did not show as much land subsidence.
- Key Words: Sentinel, Interferometry, Subsidence, Groundwater, Land Cover

#### 1 1. Introduction

- Groundwater abstraction has a variety of effects on the environment. The hazardous consequences of excessive groundwater
- abstraction include loss of ecosystems (Whiteman et al., 2010; Eggleston & Pope, 2013), land subsidence (Carruth & Conway,
- 2015), and subsequent damage to structures (Samsonov et al., 2016). Climate change, imbalance in aquifer recharge and the
- absence of groundwater abstraction regulations further exacerbate these geo-environmental hazards (Eckhardt & Ulbrich., 2003;
- Khan et al., 2013; Furuno et al., 2015). Various methods are employed to measure land subsidence due to excessive use of
- groundwater such as traditional leveling, extensioneter, numerical modeling and global positioning system (GPS) (Huang et al.,
- 2012; Hung et al., 2012; Sato et al, 2003; Zhu et al., 2015; Shearer, 1998). Measurement based on these conventional methods are
- generally restricted to a small spatial extent, so they often fail to portray a spatial representation of the land subsidence due to
- groundwater use for which spatial extent can exceed tens of kilometers (Zhu et al., 2015).
- The satellite-based Interferometric Synthetic Aperture Radar (InSAR) technique can resolve these problems. The InSAR technique
- measures the land subsidence based on multiple SAR images taken from slightly different satellite position (Massonnet & Feigl.,
- 1998). Since the swath width of satellite SAR images spans several hundred kilometers and the sensors of satellite SAR instrument
- have a very high precision to measure the radar signal phase, the satellite-based InSAR technique is capable of monitoring large
- scale land surface deformation up to the precision of a few millimeters (Rosi et al., 2016; Rosen et al., 2000 and Gabriel et al.,
- 1989). Despite these merits, the early stage SAR satellites came with some limitations such as an extensive waiting time needed
- for the acquisition, processing and delivery of the data. Moreover, their coverage is mostly restricted to well-defined locations and
- it is not always technically and economically feasible to order data acquisition of a desired target location (Rucci et al., 2012).
- Nevertheless, the satellite-based InSAR technique has been applied to monitor various aspects of land surface deformation across
- the world (Galloway and Burbey, 2011). Persistent Scatterer (PS) and Small Base Line Subset (SBAS) techniques of InSAR have
- been used to monitor land subsidence due to excessive groundwater extraction (Motagh et al., 2017; Samsonov, et al. 2016;
- Gheorghe & Armas., 2016; Zhu et al., 2015; Liu et al., 2011; Osmanoğlu et al., 2011) and other fluids for geothermal power plants
- (Sarychikhina et al., 2011), identification of contribution to land subsidence from tectonic, hydrogeological and anthropogenic
- process (Huang et al., 2016, Huang et al., 2006), charactering the effects of geological structure (fault lines) on land subsidence
- (Fruneau et al., 2001; Amelung et al., 1999) and land subsidence monitoring for the mitigation of future impacts and groundwater
- resource management (Castellazzi et al., 2016).
- Furthermore, the Sentinel-1 SAR mission can further reduce the aforementioned limitations of the earlier satellite SAR missions.
- This mission operates two satellites that constantly take the SAR imageries and has the following merit compared to its
- predecessors: (1) it can potentially provide the SAR imagery of the same spatial target every six days, which is a significant
- improvement in temporal resolution (Rucci et al., 2012); (2) it has the spatial coverage encompassing the entire globe; (3) it has
- the improved spatial and radiometric resolution (5 m x 20 m); and (4) the data processing time between the image acquisition and
- the free data dissemination to the public can be as little as 24 hours (Torres et al., 2012).
- Because of the relatively recent deployment, only a few studies have been carried out to detect the land deformation based on
- Sentinel-1 InSAR analysis. Mora et al. (2016) and Wang et al. (2017) employed the Sentinel-1 InSAR technique for the mapping
- of surface deformation due to earthquake in Japan and China, respectively. Other examples of Sentinel-1 InSAR application include

- the analysis on urban development monitoring in Spain (Bakon et al. 2016) and the study performed for the confirmation and
- comparison of historical InSAR results of land subsidence in Mexico (Sowter et al., 2016).
- Taken these together, the major contributions of this study are as follow: (1) No study has yet extensively investigated the spatial
- and temporal pattern of land subsidence due to excessive ground water use using ESA Sentinel satellite data. For this purpose, we
- first analyzed a time series of Sentinel-1 interferograms for a period of 4 seasons x 2 years. Then, we tried to relate the spatial
- pattern of the land subsidence to that of the land cover, which is a good indicator of groundwater use of the study area where
- surface water barely exist due to very low rainfall. Here, the land cover of the area was also derived from the Sentinel-2
- multispectral image; (2) While many InSAR studies deal with the subsidence in urban and developed areas because of the greater
  availability of PS points and hence, the accuracy of the InSAR analysis is higher for these areas, this study tried to obtain the land
- subsidence values of the area composed of not only the urban settings but also the agricultural and barren area. We overcame the
- limited accuracy of the InSAR analysis in these areas by applying the image filter of the radar amplitude dispersion index obtained
- from analyzing the stack of SAR images used in this study.

#### 14 2. Data and Methodology

#### 15 2.1. Data Description

16 For interferometry analysis to detect land subsidence, we used twenty nine single look complex (SLC) processing level (L-1) SAR 17 images of the study area that were provided by the European Space Agency (ESA) through Copernicus Open Access Hub 18 (https://scihub.copernicus.eu/). The images were acquired by the C band SAR sensor (wave length 5.56 cm) onboard the ESA 19 Sentinel-1A satellite with a spatial resolution of 5 m x 20 m (range x azimuth). Each image was acquired along the descending 20 track direction in interferometric wide (IW) swath mode in VV polarization, the SLC images contain 3 bands comprising complex 21 phase information and the radar intensity and(or) amplitude. The Sentinel-1 IW mode makes use of the Terrain Observation and 22 Progressive Scan (TOPS) technique (De Zan, & Guarnieri, 2006) to minimize image scalloping and reduce noise. Due to the 23 operating principle of TOPS, each IW-SLC product is inherently composed of three sub swaths designated as IW1 through IW3. 24 Each sub swath is further composed of a series of bursts demarcated by black lines as shown in illustration "a" of Fig. 1, the sub 25 swaths and bursts are overlapped for a few meters so that a seamless tile is obtained after image processing. The detail of images 26 used in this study is shown in Table 1, while the image processing flow diagram is shown in Fig. 1. 27

28

#### Table 1

#### 29 2.2. Sentinel-1 InSAR Analysis

30 We used the ESA's Sentinel-1 tool box of the SentiNel Application Platform (ESA-SNAP, 2017) for the processing of IW-SLC

- 31 images of the study area. Figure 1 describes the process of interferometric analysis: (a) the IW-SLC-L1 products are first split into
- 32 individual IW1 and IW2 because our study area lies in these two sub swaths as shown in Fig. 2; (b) the Sentinel-1 precise orbit
- files are applied to the IWs to determine accurate satellite position; (c) The sub swaths are then de-burst to remove the demarcation;

1 (d) the IWs are merged together to obtain a seamless tile for each image; (e) to reduce the processing time during further analysis, 2 a subset image of the Area of Interest (AOI) was created using the extents of study area polygon. In the next step (f) for the 3 interferogram formation, the slave images are resampled with respect to the spatial domain of the master image so that each 4 interferogram precisely represents the same spatial location, following the co-registration n-1 number of interferograms are created, 5 (g) the SRTM 3 arc seconds digital elevation model was used to correct the interferograms for topographic phase; Then, (h) an 6 image filter is applied to increase the signal to noise ratio to increase the accuracy during the unwrapping process (Goldstein & 7 Werner, 1998); Then, (i) the persistent scattering (Ferretti et al., 2000; Hooper et al., 2007) and unwrapping of the interferogram 8 phase (Chen & Zebker, 2000; Chen & Zebker, 2001; Chen & Zebker, 2002), are performed to obtain the land subsidence value in 9 the unit of millimeters at each pixel of the image. The Amplitude Dispersion Index (ADI) filter (Ferretti et al., 2001) that is applied 10 in this step helps excluding the low-accuracy pixels form the analysis. The ADI value, which is calculated for each pixel of the 11 image, represents the ratio of the standard deviation to the mean of the radar amplitude observations calculated across the stack of 12 selected images. We adopted a threshold value of 0.25 of the ADI index which is widely recommended and used to identify the 13 Persistent Scattering (PS) pixels (Ferretti et al., 2001; Liu et al., 2011). Then, the land subsidence values at the identified PS pixels 14 were interpolated using the Ordinary Kriging method to obtain the map of land subsidence. Lastly, (j) the ESA Sentinel-2 15 Multispectral Imager (S-2 MSI) product which has a spatial resolution of 10 m was used to generate a land cover map of the study 16 area. The image was classified into the following land cover classes specific to the study area using the maximum likelihood 17 algorithm: i) Barren land, ii) vegetation (orchard), iii) vegetation (seasonal), iv) urban area, and v) water. The land cover 18 classification map was resampled at the resolution of the land subsidence map and finally the accumulated subsidence at different

19 land cover classes was compared.

20

#### 21 2.3 Groundwater use in the study area

22 The Quetta valley is a part of Pishin Lora basin located in the South West province of Baluchistan, Pakistan. The valley is 23 surrounded by the Murdar Mountain to the east and Chiltan to the west (Umar et al., 2014) as shown in Fig. 2. The area is 24 climatically arid with mean annual precipitation of 200 mm and mean minimum and maximum temperature of 9 and 28oC 25 respectively. The area is located at an average elevation of 1650 m above MSL with mean annual evapotranspiration rate 1750 26 mm, which is greater than the mean annual precipitation due to the agricultural activity supplemented by groundwater irrigation. 27 Quetta valley is underlain by two main types of aquifers, the top most layer which forms the valley floor is the unconfined alluvial 28 aquifer and is composed of a mixture of gravel, sand and clay, and its depth varies between 200 to a maximum of 1000 m at certain 29 points (Alam & Ahmad, 2014). The second type of aquifer, which extends from 200+ to 1500 m is a confined aquifer and is 30 composed of limestone formation. The confined aquifer is recharged in the piedmont region of the surrounding hills (Khan et al., 31 2013). The valley is historically famous for growing high value fruits where groundwater is the main source of moisture to 32 supplement plant growth. During the 1970s, irrigation was mainly practiced through dug wells and karez system (horizontal tunnel 33 meeting with groundwater table). Since then the population of the valley has tremendously increased mainly due to increased 34 economic activities and continues influx of war refugees from Afghanistan. Groundwater use for irrigation and domestic supply 35 increased sharply during the 1980s due to the introduction of electric pumps in the area and expansion of transportation networks

1 to large markets which left many of the dug wells and the karez system dry (van Steenbergen et al., 2015). Eventually to cope with 2 the increasing water demands, high capacity electric pumps started to replace the remaining dug wells and karez systems, which 3 further exacerbate the situation. During the 1990s the groundwater table fell down to a depth of 80 m below the surface and 4 pumping became financially infeasible for poor farmers, which compels them to switch to non-agricultural based livelihoods. The 5 worsening groundwater availability was further affected during the 2003-4 drought in the country, which triggered the in-depth 6 investigation of Baluchistan Irrigation and Power Department (BIPD) to reveal that the annual amount of groundwater abstraction 7 was 97.65 million cubic meter with only 62 percent (61.15 million cubic meter) of the this amount being recharged (Khan et al., 8 2013). For many years several studies have revealed significant decline in water table ranging from 0.25 to 1.5 m year<sup>-1</sup> (WAPDA, 9 2001; Khan et al., 2013; Alam & Ahmad, 2014) but unfortunately, no solid regulation on groundwater use was imposed during all 10 this period of resource depletion except a restriction on new electricity connection for agricultural tube wells in 2008. Eventually 11 the worst impacts of excessive exploitation of groundwater have started to emerge in the form of land subsidence. The first instance 12 of significant land subsidence was observed during 2011 with the appearance of 10 m deep ground cracks on an 8 km long and 2 13 km wide swath which damaged houses, roads and a hospital (Khan et al., 2013; Kakar et al., 2016). Current estimates show that 14 there are around 11,000 pumping wells in the area mostly owned by individuals (BDS, 2013) but the exact number and locations 15 of active wells are unknown except those owned by the government agencies as shown in Fig. 2. Majority of these wells are now 16 located in the urban and agricultural area. 17 18 Figure 2.

19

#### 20 3. Results and Discussions

### 21 3.1. Distribution of Persistent Scattering (PS) Points

22 Figure 3 compares the map of the ADI value and land cover. The ADI map was generated using the time series amplitude bands 23 of the SAR images. In general, the ADI value is high in the urban and barren area because of the existence of many manmade 24 structures and rocks outcrop, respectively. However, the areas occupied by plants, which is classified as orchard and seasonal 25 vegetation has low ADI value. This is particularly because the fluttering of small plant leaves due to various meteorological reasons 26 such as winds and rain can disturb consistent reflectance (Liu et al., 2011). Therefore, using the land subsidence values of all pixels 27 in each land classification class can cause significant inaccuracy in the resultant land subsidence map. To resolve this issue, the 28 subsidence identified at the "Persistent Scattering (PS) points" at which ADI value is lower than 0.25 were first selected for further 29 analysis. Then, the land subsidence values at these PS points were interpolated using the Ordinary Kriging technique to obtain the 30 map of the land subsidence. Figure 4 shows the empirical cumulative distribution function of the ADI values in the four major 31 land cover classifications in the study area. At an ADI threshold of 0.25, approximately 35%, 30%, 22%, and 18% of barren land, 32 urban area, seasonal vegetation, and orchard were identified as the PS points, respectively.

| 1 | Figure 3. |
|---|-----------|
| 2 |           |
| 3 | Figure 4. |

#### 4 **3.2. Land subsidence**

#### 5 3.2.1. Distribution of land subsidence

6 We analyzed a total of twenty eight interferograms to monitor land subsidence in the study area covering a period of two years. 7 Figure 5 shows the normalized histogram of the land subsidence values of the eight selected interferograms (4 seasons x 2 years = 8 8), the sub-titles are shown in YYYYMMDD format. The subsidence value shown on the x-axis of each of the plots represents the 9 amount of subsidence from its original position detected on the reference date (20141016), the y-axis shows the relative frequency 10 of subsidence normalized by total number of pixels. In Fig. 5 (a) to (c) a gradual leftward movement of histogram bar masses 11 from 40-60 mm implies a steady increase in the magnitude of overall subsidence. The bimodal pattern of the histograms after (d) 12 20151104 implies that the subsidence at different locations is occurring at different rates. Figure 5(h) shows that more than 100-13 120 mm of subsidence has occurred on the average in the study area for the period of 2 years. Some area experienced more than 14 200 mm of subsidence for the same period. Considering that subsidence is primarily related to the groundwater use which varies 15 greatly according to land use and aquifer compressibility, this phenomenon is likely to have been caused by the combination of 16 spatially unbalanced groundwater extraction and heterogeneous aquifer compressibility. We will discuss this matter in detail 17 through Fig. 6 and 7. 18 Figure 5.

#### 19 3.2.2. Maps of land subsidence

20 Figure 6 shows the subsidence maps generated by interpolating the subsidence values at the PS points. The color scale varies 21 between 60 mm and 240 mm, and the same color scale applies to all maps. With the progress of time, land subsidence gradually 22 expands across the entire spatial extent of the study area. There are mainly two distinct pattern of subsidence, the first pattern with 23 subsidence ranging from 100 mm to more than 240 mm is particular to the urban area of Quetta. This area of accelerated subsidence 24 is denoted as blue colors in each of the subsidence maps in Fig. 6. The expanding blue region indicates that the subsidence is 25 propagating to the surrounding areas with the passage of time. The second pattern which ranges from 60 mm to 120 mm is prevalent 26 over the rest of the study area and is denoted as red to yellow color. In addition, it is interesting to note that the subsidence in the 27 north and south-western barren land Fig. 6d has recovered in Fig. 6e, this effect will be explained with the help of Fig. 9. Overall 28 our result shows a subsidence rate of 30-120 mm year<sup>-1</sup> which is in very close comparison with the subsidence rate of 81-116 mm 29 year-1 and 29-120 mm year-1 determined by GPS stations in the study area independently (Khan et al., 2013 and Kakar et al., 2016). 30 31 Figure 6.

## 1

#### 2 3.2.3. Relationship between land subsidence and land cover

3 By comparing a series of subsidence maps of Fig. 6 and the land cover map (Fig. 3b), it can be noted that the subsidence has strong 4 correlation with land cover types. The subsidence at urban and vegetated area (primarily occupied by orchard farming) is 5 significantly greater than that of the seasonal vegetation or barren land. In addition, the higher subsidence in the southern part of 6 the study area also coincides with the agricultural area. Considering that evapotranspiration is significantly greater than 7 precipitation and that there is no sizeable surface water source in the study area, it is a solid assumption that most of the agricultural 8 and urban activities are supplied by the extracted groundwater leading to land subsidence, and the result of the interferometry 9 analysis of this study captures this tendency very well. The accumulated subsidence map shown in Fig. 6 (h) 20161005 was further 10 analyzed by filtering the subsidence values according to four major land cover classes in the study area. The result is shown in Fig. 11 7. This analysis shows that subsidence within the range less than 200 mm during the two-year period is mostly recorded in the 12 barren and seasonally cultivated areas as shown in Fig. 7 (a) and (b), respectively. On the contrary, large subsidence values are 13 mostly recorded in the urban and vegetated area as shown in Fig. 7 (c) and (d) with a considerable proportion of subsidence pixels 14 in excess of 200 mm. It is also likely that the few pixels which showed subsidence in excess of 200 mm in barren and seasonally 15 cultivated land were influence by the large subsidence in urban and vegetated (orchard) areas as the land covers are closely meshed 16 together as shown in Fig. 3 (b). 17

18

19

#### 20 3.2.4. Time series analysis of land subsidence for major land cover types

21 Figure 8 shows the time series comparison of land subsidence for four major land use types using all twenty eight interferograms. 22 The shaded paths of subsidence for each land cover type represent the interquartile range of land subsidence while the solid lines 23 represent the average subsidence. The range of subsidence varies with time and comparatively barren land cover has the least while 24 the vegetated land under orchard has the highest range of subsidence. For all land cover types the least value of subsidence range 25 is observed during March to May 2015 which is also the period of consistently higher precipitation. 26

Figure 7.

#### Figure 8.

Figure 9 compares the average subsidence of four land cover types with the subsidence rate measured at two GPS stations as 30 reported by Kakar et al., 2016. The GPS station 1 (SBKW) shows an overall subsidence rate close to that of our study while the

subsidence rate at GPS station 2 (Jhak) is comparatively higher. This difference in subsidence rate could be due to two reasons i.e.

i) the fact that the GPS station characterizes a point measurement which lacks to capture the aggregated spatial subsidence pattern,

- ii) the frequency of measurement at the GPS stations is very low as compared to our study. In addition, it is interesting to note that
- the recovery of temporary subsidence during the interval 20151104-20160115 in barren land is due to soil dilation which matches
- with the end of annual prolonged dry spell in the study area, assuming that there is no human intervention in the barren land and it
- does not receive any irrigation therefore the extreme soil shrink-swell cycle due to soil moisture change may have caused this
- temporary effect in barren land. The vegetation land cover also shows a great degree of similar effect but that could not be attributed
- to soil dilation alone as there are frequent human intervention in the form of land plowing and cultivation operations. Overall our
- result of land subsidence is well within the GPS measured range

#### Figure 9.

#### 10 4. Conclusion

- InSAR technique was employed to analyze land subsidence at Quetta Valley, Pakistan for the two-year periods between October
- 16, 2014 and October 5 2016. Twenty eight pairs of the Sentinel-1 SAR images were used as the input data of the analysis. Over
- the two year period, the entire Quetta valley area experienced severe land subsidence exceeding 250 mm at some locations. We
- compared the map of the land subsidence to the land cover types of the study area by analyzing the Sentinel-2 multispectral images.
- The result revealed that the subsidence was more severe in urban and agricultural area where groundwater extraction is significant.
- Barren and seasonally cultivated area experienced less severe subsidence. Considering that groundwater use is expected to increase
- in the area due to increasing populations, the area will experience more land subsidence, which calls for immediate regulatory
- action to alleviate the damages to the man-made structures and the environment.
- We found through this study that the ESA-Sentinel satellite data can be a very useful tool to identify spatio-temporal evolution of
- land subsidence across the study area varying with diverse land cover types, which could be a daunting task according to traditional
- approaches based on point measurements or leveling. No financial cost was invested to obtain the satellite data and the data
- processing software tools because they are provided free of cost by the ESA. We expect our methodology can be applied to other
- locations across the world to resolve critical issues of land subsidence and groundwater resources management.

#### 24 Acknowledgment

- This research was fully supported by a grant [MPSS-NH-2015-79] through the Disaster and Safety Management Institute funded
- by Ministry of Public Safety and Security of Korean government.

4

10

13

23

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
