# Peer review of "Detection of Land Subsidence due to Excessive Groundwater Use Varying"

_Natural Hazards and Earth System Sciences, 2017_

## Referee Comment (RC1) · Anonymous Referee #1 · 22 Jul 2017

As InSAR method has been widely and maturely applied in land subsidence monitoring and high precision time-series can be easily achieved for different regions. Numerous publications show good performance of this kind of application.

For this specific test study, the precision of InSAR results is not high enough to show the spatial distribution and temporal evolution characteristics, as compared with GPS in Figure 9. Besides, the quantitative correlation between surface subsidence and the underground water fluctuation is missed, i.e. the well data should be involved to show the inner correlation between land subsidence and the ground water changes.

[Figure]

Some detailed techniques such as atmospheric effect, noise filter and external DEM (SRTM 3 arc-sec. rather than 1 arc- sec) are lack or not updated, which makes the InSAR results be not good.

Which strategy (InSAR processing method) is applied in this study, as the description of the InSAR technique is not consistent.

Some typo and spelling error should corrected.

---

## Referee Comment (RC2) · Anonymous Referee #2 · 15 Aug 2017

Generally comments

This paper analyzed the evolution of land subsidence from Oct. 2014 to Nov. 2016, and subsidence features under different land cover types in Pakistan. During the process Sentinel satellite data were adopted. This study is interesting.

Specific comments

1.This paper worked on detecting land subsidence varying with land cover types. The accuracy of land cover classification is very important. What is the accuracy? Do you

take some validation work? Please give the detail.

2.Where is the reference point during the SAR data process? Is the subsidence value at reference point equal to zero? However, paragraph 3.2.2 shows that subsidence rate is 30-120 mm/year in the study area. Do you make some calibration considering the reference point?

3.Generally PS points in vegetated areas are rare. How do you guarantee the accuracy of land subsidence in these areas?

4.What is the elevation range in study area? Can the SRTM 3 arc seconds digital elevation model with a resolution of 90 m meet the demand for topographic phase?

Technical corrections

1. The legend may be wrong in Figure 6. The dark blue should represent < -241, not <241.
* * *

---

## Referee Comment (RC3) · Anonymous Referee #3 · 20 Aug 2017

General comments

The manuscript presents a detection of land subsidence in different land cover types in Quetta valley, Pakistan based on ESA Sentinel satellite data. The analysis is helpful to explain the impact of human activities (excessive groundwater use) on land subsidence. However, there are still some important issues that need further explanation.

Specific comments

1. The authors attempt to analysis land subsidence due to excessive groundwater

use in order to address the impact of excessive groundwater use on land subsidence. However, the whole manuscript only presents the land subsidence in different land cover types rather than exploitation of groundwater. And there is also no quantitative relationship between different land cover types and the groundwater use. It is the main research topic and should be clearly explained.

2. The errors of the Sentinel-1 InSAR data for detection land subsidence and the comparison with other data should be presented.

3. Why the author used the SRTM 3 arc seconds DEM rather than SRTM 1 arc seconds DEM, which is more accuracy and also free.

4. The author applied the Amplitude Dispersion Index (ADI) filter to exclude the low-accuracy pixels. Why the threshold value of 0.25 of the ADI index is selected? Which low-accuracy pixels are exclude need to be presented in a spatial distribution map together with the high-accuracy pixels? The ratio of low and high accuracy pixels, the precision corresponding to different thresholds?

5. Why is the Ordinary Kriging method applied to obtain the map of land subsidence rather than other interpolation methods?

6. There are twenty eight pairs of the Sentinel-1 SAR images in the study area. Why the eight dates in Figure 5 and Figure 6 is selected to analysis?

7. The subsidence at urban area shows not so significantly greater than that of the seasonal vegetation or barren land in Figure 7. Are there significant differences between the land subsidence in four land cover types in statistics?

8. The acquisition time of the ESA Sentinel-2 data used to the detection of land cover classes should be presented. The original image should be mapped. The precision of the detection of land cover types also need to presented.

Technical corrections

1. The right parentheses is absence in the title of Figure 5.

2. Page 4 line 24, the Celsius degree should be revised.

3. The resolution of figures need to be improved.

---

## Author Comment (AC1) · 23 Sep 2017

\*\* Please find the attachment for the original response document with the figures and tables. You can find the link to this original response document at the end of this document.

Referee#1

We truly appreciate your review on our manuscript. All of your comments were extremely helpful for us to learn more about this topic and to improve the quality of the

article. Please find below the responses to your specific comments.

1. Your comment: As InSAR method has been widely and maturely applied in land subsidence monitoring and high precision time-series can be easily achieved for different regions. Numerous publications show good performance of this kind of application. For this specific test study, the precision of InSAR results is not high enough to show the spatial distribution and temporal evolution characteristics, as compared with GPS in Figure 9.

Our response: As you indicated, our InSAR subsidence does not precisely match the GPS-based measurement as shown in Figure 9. We found that this issue is primarily caused because our InSAR subsidence shown in Figure 9 represents the average of all subsidence pixels classified as a given land cover while the GPS-based measurement represents the subsidence at a given point location. Figure S-1 compares the subsidence trend based on the GPS measurement and our InSAR subsidence at the pixel including the GPS point. The plot shows the improved match with the R2 value of 0.94.   2. Your comment: The quantitative correlation between surface subsidence and the underground water fluctuation is missed, i.e. the well data should be involved to show the inner correlation between land subsidence and the ground water changes.

Our Response: As you pointed out, our manuscript did not investigate the relationship between the surface subsidence and groundwater fluctuation due to the difficulty of acquiring groundwater data. Following your specific comment, we acquired the groundwater data of the study area from the provincial irrigation department. The groundwater level was recorded every 1.5∼2 month, so we collected a total of 70 observations (14 instances during the 2 years period x 5 locations in the study area as shown in Figure S-2). The date of groundwater observation did not always coincide with the SAR image acquisition date, so we linearly interpolated the groundwater level for the day of SAR image acquisition. Figure S-3 (a) compares the change in groundwater level (x) and our InSAR subsidence (y) of the same date. The plot reveals two distinct clusters of linear trend between the two variables. The first cluster of mild linear trend ranging from

0.029 to 0.041 (mm/mm) (purple and blue lines) was found towards the eastern part of the Quetta valley where urbanization is concentrated with large changes in groundwater level (Kakar et al., 2016). The second cluster of the linear trend ranging from 0.059 to 0.11 (mm/mm) (green, yellow and orange line) was identified in the opposite side of the Quetta valley where farming activity is severe. Figure S-3(b) shows the temporal trends of groundwater decline at the five selected locations with a decline of 3100 and 3400 mm/year at the Killi Nasran and Killi Shabak, respectively; while the mild slope of groundwater decline ranging from 985 to 1530 mm/year was observed at the remaining three locations. The groundwater use throughout the area is extensive due to the fact that there is no significant alternate water source, but still the decline in groundwater levels is characterized by a maximum and minimum rate. Similar results of contrasting trends of groundwater decline in the east west direction have been found earlier by Khan et al. (2013) and Kakar et al. (2016). One reason for the mild decline of groundwater in the western side of Quetta valley could be the natural recharge of groundwater through the agricultural lands. Secondly, this is a widely reported fact that the irrigation methods and its efficiencies in this region are among the lowest i.e. 40 to 60 percent due to poor maintenance of the irrigation infrastructure (Khair et al., 2010), hence excessive percolation from the irrigation system could also contribute to the artificial recharge or recycling of groundwater in the western part. Furthermore, Oldham & Pascoe (1939) have identified the existence of an impermeable vertical layer of clay up to 7 km in length and up to 3 km wide in the north south direction through the center of Quetta valley. This natural barrier is believed to hydraulically isolate the two sides and hence the groundwater decline rates are different in the east west direction. While this result is not comprehensive enough to prove the groundwater-subsidence relationship across the entire study area due to the complex geology and the lack of groundwater observation points, we believe that this result partly proves that there has been spatially heterogeneous ground water decline in the study area that is strongly dependent on the degree of human activity and that this spatial heterogeneity is reflected through the land cover based subsidence as we suggested in this article.

3. Your comment: Some detailed techniques such as atmospheric effect, noise filter and external DEM (SRTM 3 arc-sec. rather than 1 arc- sec) are lack or not updated, which makes the InSAR results be not good.

Our Response: We already applied the Goldstein filter (line 6, page 4) and the external DEM (3 arc sec) (line 5, page 4) to improve the accuracy of the InSAR subsidence values, but we did not perform the correction for atmospheric effect. Following your suggestion, we performed the correction for the atmospheric effect. We used the atmospheric phase delay model of Baby et al., (1988) in combination with climatic data ERA Interim from the European Center for Medium-Range Weather Forecasts (ECMWF) to generate atmospheric delay differential maps in accordance with the SAR interferograms. Figure 4(a) shows the two contrasting cases of subsidence before and after the atmospheric effect across the study area, the differential maps are for the date of 2014-10-16 and 2014-11-09. Figure S-4(b) shows the amount of atmospheric correction (mm) at the GPS location of Figure S-2. From the mild to severe cases, the subsidence was corrected by -0.59 mm through 49.95 mm at the GPS point. The effect was more pronounced during the summer months as compared to the winter months. After the correction, the time series of land subsidence showed less fluctuation as shown in Figure S-4(c).  

Lastly, we agree to incorporate all your constructive comments and to clearly describe the InSAR processing method as differential DInSAR, and the correction of typo and spelling errors in the revised manuscript.

Additional references

Baby, H. B., Gole, P., & Lavergnat, J. (1988). A model for the tropospheric excess path length of radio waves from surface meteorological measurements. Radio science, 23(06), 1023-1038.

Kakar, N., Kakr, D. M., Khan A. S., Khan, S. D. (2016). Land Subsidence Caused by Groundwater Exploitation in Quetta Valley, 32 Pakistan. Int. j. econ. environ. geol.

Vol:7(2) 10-19.

Khan, A. S., Khan, S. D., & Kakar, D. M. (2013). Land subsidence and declining water resources in Quetta Valley, Pakistan. Environmental earth sciences, 70(6), 2719-2727.

Khair, S. M., Culas, R. J., & Hafeez, M. (2010). The causes of groundwater decline in upland Balochistan region of Pakistan: implication for water management policies. In Australian Conference of Economists (ACE10), Sydney, Australia.

Oldham, R. D., & Pascoe, E. H. (1939). A Manual of the Geology of India and Burma. Order Of The Government Of India.

Referee #2

We truly appreciate your review on our manuscript. All of your comments were extremely helpful for us to learn more about this topic and to improve the quality of the article. Please find below the responses to your specific comments.

1. Your comment: This paper worked on detecting land subsidence varying with land cover types. The accuracy of land cover classification is very important. What is the accuracy? Do you take some validation work? Please give the detail.

Our Response: As you suggested, the accuracy of the land cover map shown in Figure 3(b) should be addressed. We will add a corresponding section in the article as follow: We used the bands 2 to 8, 8A, 11 and 12 of the Sentinel-2 multispectral image to classify the study area into 6 land cover classes. We used the method of error matrix analysis in combination with Kappa coefficient (Gomez & Montero, 2011) to assess the accuracy of land cover classification. The analysis result revealed that the overall accuracy of the land cover classification is 81 % with the Kappa coefficient of 0.76. Table S-1 shows the error matrix.

2. Your comment: Where is the reference point during the SAR data process? Is the subsidence value at reference point equal to zero? However, paragraph 3.2.2 shows that subsidence rate is 30-120 mm/year in the study area. Do you make some

calibration considering the reference point?

Our Response:

We selected the SAR image of 16 Oct 2014 as the reference for comparison where the subsidence is taken as zero and then subsequently compared the subsidence from the following interferogram pairs to this point. The value of 30-120 mm/year subsidence is the overall average range of subsidence with regard to 16 Oct 2014 in the study area during the 2 years period (minimum and maximum accumulated subsidence during the 2 years divided by the time, from Figure 6). For the issue of calibration/comparison of InSAR subsidence with the reference point we have updated this deficiency by extracting the InSAR subsidence at the GPS location from the interferograms and compare this value with the GPS reading. Figure S-1 shows this comparison, which indicates the good match between the InSAR-based and GPS-based subsidence with the R2 value of 0.94.

3. Your comment: Generally PS points in vegetated areas are rare. How do you guarantee the accuracy of land subsidence in these areas?

Our Response:

We agree. The density of PS points in vegetated area is low as shown in Figure 4 of the paper. This problem is caused because the reflection of the SAR with a short wave length is particularly sensitive to the densely vegetated areas. Therefore, this problem should be addressed in two folds: the wave length of the SAR and the vegetation density. Regarding the wave length of the SAR, the wave length of the Sentinel-1 C band that was used in this study is 56 mm. This value is significantly greater than the wave length of other typical SAR sensors onboard satellites (e.g. 31mm for X-band), so the corresponding errors may be less for this study than the ones based on the X-band SAR imageries. Regarding the density of vegetation, this study has two different vegetation types which is seasonal and orchard vegetation. In the area of the seasonal vegetation, farmers usually do not cultivate the whole land due to the scarcity of surface
and ground water and arid climate, so the half of the area remains as unsown/fallow which can provide a potential site for PS points during the course of time. In the area of orchard vegetation, the proportion of the orchards trees to the whole area is fairly low (Figure S-5), which provide sufficient area of bare soil to act as a potential PS point.

In addition, we tried to reduce the errors associated with PS point density by applying the threshold of 0.25 of the Amplitude Dispersion Index (ADI), which is calculated across the stack of SAR images for the selection of these PS points (mentioned at page 4 line 10-13).

4. Your comment: What is the elevation range in study area? Can the SRTM 3 arc seconds digital elevation model with a resolution of 90 m meet the demand for topographic phase?

Our Response:

The elevation range in the most affected area by subsidence is 50 m while the overall elevation range in the whole of study area is 1907 m. We choose the SRTM 3 arc sec digital elevation model for the removal of topographic phase, based on the recommendation in the literature (Scaioni, 2015; Mathew et al., 2014). We also found that the complete void filled 1 arc sec DEM is only available for the United States, for the rest of the globe the 1 arc sec void filled DEM is derivative of the former (https://lta.cr.usgs.gov/SRTM1Arc). For the future research, we would like to investigate into the accuracy comparison of land subsidence with the two DEMs

The technical correction in the legend of Figure 6 has already been taken care of in the discussion paper submitted to the journal, we also agree to incorporate your constructive comments and the additional citations we referred in this response in the revised manuscript following the editor's decision.

Additional References

Gomez, D., & Montero, J. (2011, August). Determining the accuracy in image supervised classification problems. In Proceedings of the 7th conference of the European Society for Fuzzy Logic and Technology (pp. 342-349). Atlantis Press.

Mathew, J., Majumdar, R., & Kumar, K. V. (2014). Estimating the atmospheric phase delay for quantifying co-seismic deformation using repeat pass Differential SAR Interferometry: Observations from 20th April 2013 Lushan (China) Earthquake. The International Archives of Photogrammetry, Remote Sensing and Spatial Information Sciences, 40(8), 57.

Scaioni, M. (Ed.). (2015). Modern technologies for landslide monitoring and prediction. Springer.

Referee #3

We truly appreciate your review on our manuscript. All of your comments were extremely helpful for us to learn more about this topic and to improve the quality of the article. Please find below the responses to your specific comments.

1. Your comment: The authors attempt to analysis land subsidence due to excessive groundwater use in order to address the impact of excessive groundwater use on land subsidence. However, the whole manuscript only presents the land subsidence in different land cover types rather than exploitation of groundwater. And there is also no quantitative relationship between different land cover types and the groundwater use. It is the main research topic and should be clearly explained.

Our Response:

We agree. We addressed the relationship of land subsidence with groundwater level change in response to anonymous Referee #1 comment #2. Please see: Page 1 line 14-19 and page 2 line 11-25 of this document.

2. Your comment: The errors of the Sentinel-1 InSAR data for detection land subsidence and the comparison with other data should be presented.

Our Response:

We agree. This point has been also addressed in response to anonymous referee #1 comment 1 and comment 3. Please see: Page 1 line 14-19 and page 3 line 20-21 of this document.

3. Your comment: Why the author used the SRTM 3 arc seconds DEM rather than SRTM 1 arc seconds DEM, which is more accuracy and also free.

Our Response:

We choose the SRTM 3 arc sec digital elevation model for the removal of topographic phase, based on the recommendation in the literature (Scaioni, 2015; Mathew et al., 2014). We also found that the complete void filled 1 arc sec DEM is only available for the United States, for the rest of the globe the 1 arc sec void filled DEM is derivative of the former (https://lta.cr.usgs.gov/SRTM1Arc). For the future research, we would like to investigate into the accuracy comparison of land subsidence with the two DEMs

4. Your comment: The author applied the Amplitude Dispersion Index (ADI) filter to exclude the low accuracy pixels. Why the threshold value of 0.25 of the ADI index is selected? Which low-accuracy pixels are exclude need to be presented in a spatial distribution map together with the high-accuracy pixels? The ratio of low and high accuracy pixels, the precision corresponding to different thresholds?

Our Response:

We acknowledge that this threshold value is rather random, so we tried to share our thoughts on this matter through Figure 3(a) and Figure 4 in the paper, which shows the map and the histogram of the ADI filter values. However, in the end we had to choose one value to acquire the result. For the future research, we would like to investigate into this matter in another study area where in-situ land subsidence data is rich.

5. Your comment: Why is the Ordinary Kriging method applied to obtain the map of land subsidence rather than other interpolation methods?

Our Response:

We choose to use Kriging method to obtain maps of land subsidence because it is probably the most widely applied method to spatially interpolate geographical information. Many literatures use this method to interpolate land subsidence values (Béjar-Pizarro et al., 2016; Peng et al., 2002). In addition, it helps us to figure out spatial trend of the variable being interpolated and integrates in the interpolation process as well as the uncertainty of the interpolated values.

6. Your comment: There are twenty eight pairs of the Sentinel-1 SAR images in the study area. Why the eight dates in Figure 5 and Figure 6 is selected to analysis?

Our Response:

This is correct, we have 28 pairs of Sentinel-1 based interferograms in the study area. The result of all 28 pairs is shown in another format in Figure 8 and 9, where it is meant to present the temporal evolution of land subsidence. In Figure 5 and 6, for the purpose of summarizing the spatial and temporal subsidence we selected 4 seasons in a year and during the 2 years study period it gives the 8 dates each corresponding to a seasonal interferogram.

7. Your comment: The subsidence at urban area shows not so significantly greater than that of the seasonal vegetation or barren land in Figure 7. Are there significant differences between the land subsidence in four land cover types in statistics?

Our Response:

Following this suggestion, we performed the two sample K-S test to determine whether land subsidence of a given land cover type resembles one another or not. Table S-2 summarizes the test result, which reveals that the subsidence at barren and urban land cover are not significantly different from each other at 5% significance level, while the subsidence at other land covers is significantly different from each other with varying degree of significance.

8. Your comment: The acquisition time of the ESA Sentinel-2 data used to the detection of land cover classes should be presented. The original image should be mapped. The precision of the detection of land cover types also need to presented.

Our Response: The Sentinel-2 original image along with the acquisition date used in the detection of land cover classes is presented in Figure S-6. We assessed the accuracy of land cover classification using the widely used approach of error matrix (Response to comment #1 of anonymous referee #2 i.e. Table S-1 page 6 line 12-22) and the method of Kappa coefficient (Gomez & Montero, 2011). The overall accuracy of the land cover classification was found to be 81 % with Kappa coefficient of 0.76.

Additional Reference:

Béjar-Pizarro, M., Guardiola-Albert, C., García-Cárdenas, R. P., Herrera, G., Barra, A., López Molina, A., ... & García-García, R. P. (2016). Interpolation of GPS and Geological Data Using InSAR Deformation Maps: Method and Application to Land Subsidence in the Alto Guadalentín Aquifer (SE Spain). Remote Sensing, 8(11), 965.

Gomez, D., & Montero, J. (2011, August). Determining the accuracy in image supervised classification problems. In Proceedings of the 7th conference of the European Society for Fuzzy Logic and Technology (pp. 342-349). Atlantis Press.

Mathew, J., Majumdar, R., & Kumar, K. V. (2014). Estimating the atmospheric phase delay for quantifying co-seismic deformation using repeat pass Differential SAR Interferometry: Observations from 20th April 2013 Lushan (China) Earthquake. The International Archives of Photogrammetry, Remote Sensing and Spatial Information Sciences, 40(8), 57.

Peng, M. H., & Shih, T. Y. (2002). A quality assurance approach for land subsidence interpolation. Survey Review, 36(286), 568-581.

Scaioni, M. (Ed.). (2015). Modern technologies for landslide monitoring and prediction. Springer.

Please also note the supplement to this comment:
https://www.nat-hazards-earth-syst-sci-discuss.net/nhess-2017-234/nhess-2017-234-AC1-supplement.pdf

————————————————

[Figure]

Figure S-1. InSAR subsidence (y) versus GPS subsidence (x)

**Fig. 1.** InSAR subsidence (y) versus GPS subsidence (x)

[Figure]

Figure S-2. Map of the study area showing the location of groundwater observation wells, GPS
station and other important features.

**Fig. 2.** Map of the study area showing the location of groundwater observation wells, GPS
station and other important features.

[Figure]

Figure S-3(a) and (b). Quantitative correlation between land subsidence and change in groundwater
levels and temporal groundwater decline at the observation points

**Fig. 3.** Quantitative correlation between land subsidence and change in groundwater levels and temporal groundwater decline at the observation points

[Figure]

Figure S-4. Atmospheric effects and its application to the correction of InSAR land subsidence

**Fig. 4.** Atmospheric effects and its application to the correction of InSAR land subsidence

[Figure]

Figure S-5. Grid layout of orchard vegetation in the study area showing significant bare soil surface to act as a potential PS point. (Farm location: 66.9474 E, 30.1187 N, Image date: July 4, 2016)

**Fig. 5.** Grid layout of orchard vegetation in the study area showing significant bare soil surface to act as a potential PS point. (Farm location: 66.9474 E, 30.1187 N, Image date: July 4, 2016

Figure S-6. Sentinel 2 RGB color composite image of July 4, 2016 used for the classification of land cover classes

**Fig. 6.** Sentinel 2 RGB color composite image of July 4, 2016 used for the classification of land cover classes

Table S-1. Error matrix of land cover classification

|  | Barren (Hill) | Barren (High land) | Vegetation (Orchard) | Vegetation (Seasonal) | Urban Area | Water | Total | Accuracy (User) |
|---|---|---|---|---|---|---|---|---|
| Barren(Hill) | 248 | 11 | 3 | 1 | 7 | 0 | 270 | 92% |
| Barren (High land) | 7 | 129 | 8 | 9 | 11 | 0 | 164 | 79% |
| Vegetation (Orchard) | 5 | 7 | 143 | 52 | 9 | 1 | 217 | 66% |
| Vegetation (Seasonal) | 3 | 7 | 33 | 172 | 7 | 0 | 222 | 77% |
| Urban Area | 9 | 8 | 5 | 3 | 170 | 1 | 196 | 87% |
| Water | 0 | 0 | 1 | 0 | 0 | 20 | 21 | 95% |
| Total | 272 | 162 | 193 | 237 | 204 | 22 | 1090 |  |
| Accuracy (classified) | 91% | 80% | 74% | 73% | 83% | 91% |  |  |

Overall accuracy = 81 %, Kappa coefficient=0.76

**Fig. 7.** Error matrix of land cover classification

Table S-2. Kolmogorov-Smirnov test result of statistical significance of subsidence at different land cover types (0 means the subsidence is not significantly different, 1 means the subsidence is significantly different at 5% significance level, the values in parenthesis represent the p-value)

| Land Cover | Barren | Urban | Vegetation (Seasonal) | Vegetation (Orchard) |
|---|---|---|---|---|
| Barren | x | $0\ (5.9 \times 10^{-2})$ | $1\ (2.0 \times 10^{-3})$ | $1\ (9.3 \times 10^{-4})$ |
| Urban | | x | $1\ (6.0 \times 10^{-4})$ | $1\ (1.7 \times 10^{-4})$ |
| Vegetation (Seasonal) | | | x | $1\ (6.8 \times 10^{-5})$ |
| Vegetation (Orchard) | | | | x |

**Fig. 8.** Kolmogorov-Smirnov test result of statistical significance of subsidence at different land cover types